# A local ATR-dependent checkpoint pathway is activated by a site-specific replication fork block in human cells

Sana Ahmed-Seghir[1†], Manisha Jalan[1†], Helen E Grimsley[1], Aman Sharma[1], Shyam Twayana[2], Settapong T Kosiyatrakul[2], Christopher Thompson[1], Carl L Schildkraut[2]*, Simon N Powell[1,3]*

[1]Department of Radiation Oncology and the Molecular Biology Program, Memorial Sloan Kettering Cancer Center, New York, United States; [2]Department of Cell Biology, Albert Einstein College of Medicine, New York, United States; [3]Molecular Biology Program, Memorial Sloan Kettering Cancer Center, New York, United States

*For correspondence:
carl.schildkraut@einsteinmed.
edu (CLS);
powells@mskcc.org (SNP)

[†]These authors contributed equally to this work

Competing interest: The authors declare that no competing interests exist.

**Abstract** When replication forks encounter DNA lesions that cause polymerase stalling, a checkpoint pathway is activated. The ATR-dependent intra-S checkpoint pathway mediates detection and processing of sites of replication fork stalling to maintain genomic integrity. Several factors involved in the global checkpoint pathway have been identified, but the response to a single replication fork barrier (RFB) is poorly understood. We utilized the *Escherichia coli*-based Tus-*Ter* system in human MCF7 cells and showed that the Tus protein binding to *TerB* sequences creates an efficient site-specific RFB. The single fork RFB was sufficient to activate a local, but not global, ATR-dependent checkpoint response that leads to phosphorylation and accumulation of DNA damage sensor protein γH2AX, confined locally to within a kilobase of the site of stalling. These data support a model of local management of fork stalling, which allows global replication at sites other than the RFB to continue to progress without delay.

## eLife assessment

This manuscript reports **important** data on the cellular response to a single site-specific replication fork block in human MCF7 cells. **Compelling** evidence shows the efficacy of the bacterial Tus-Ter system to stall replication forks in human cells. Fork stalling led to lasting ATR-dependent phosphorylation of histone H2AX but not of ATR itself and its downstream targets RPA and CHK1.

## Introduction

Genomic DNA is constantly exposed to exogenous and endogenous damaging agents, forming DNA lesions that challenge the progression of replication forks (RFs) during the S-phase of the cell cycle. The genome contains natural impediments, which cause replication stalling such as common fragile sites, repeated sequences, non-B structures, sites of collision between transcription and replication machinery, or DNA-protein complexes (*Mirkin and Mirkin, 2007*). These replication fork barriers (RFB) can impede the progression of the replication machinery and, if unresolved, lead to genomic instability, a common hallmark of aneuploidy, neurological and neuromuscular disorders, and cancer.

Replication stress can impede the progression of the replication fork. During replication stress, the intra-S phase checkpoint pathway can be activated at a local and global level, which is a critical step to process the DNA lesions and maintaining genome integrity. The S-phase checkpoint involves the uncoupling of replicative polymerases from the replicative helicase and the generation of ssDNA,

which rapidly gets coated with RPA. In turn, this activates ATR/ATRIP kinase signaling with its effector kinase CHK1, plus phosphorylation of H2AX at Ser139 in response to a variety of lesions that promote cell cycle arrest, fork stabilization, and restart (*Iyer and Rhind, 2017*; *Iyer and Rhind, 2013*; *Willis and Rhind, 2009*; *Zeman and Cimprich, 2014*).

Previous studies suggest that the local and global intra-S checkpoints are two distinct mechanisms that can be distinguished by the signaling intensity of detecting DNA damage (*Saxena and Zou, 2022*). It is thought that a threshold of DNA damage must be reached to activate the global checkpoint and impact cell cycle progression during replication (*Bantele et al., 2019*; *Shimada et al., 2002*). The local checkpoint reacts to a unique replication fork stalling site, which does not impact overall cell cycle progression. The proteins involved in the global replication stress response have been identified using DNA damaging agents that lead to multiple DNA lesion formation randomly throughout the genome. Nevertheless, the mechanism and timing of the checkpoint pathway at a local scale remain largely unknown and are critical to understanding the series of processes at a single RF stall.

In the *Escherichia coli* genome, the DNA pausing sequences called terminator (Ter) are recognized by a protein called Tus to cause a polar site-specific arrest of the RF at the end of the bacterial chromosome replication (*Hiasa and Marians, 1994*; *Hidaka et al., 1988*; *Mulcair et al., 2006*; *Roecklein et al., 1991*). The Tus-*Ter* complex leads to a temporarily locked complex on DNA that can be overcome by the fork arriving in the opposite direction, displacing Tus to terminate replication. The artificial *E. coli*-based Tus/Ter system has previously been employed in mouse embryonic stem cells to measure homologous recombination repair products and investigate the DNA repair pathway choice (*Chandramouly et al., 2013*; *Willis et al., 2017*; *Willis et al., 2014*).

In this study, we integrated a plasmid with five repeats of the *TerB* sequence in the non-permissive orientation, referred to as pWB15, at a unique site within chromosome 12 of the breast cancer cell line MCF7 (MCF7 5C-TerB clone). We utilized the integrated Tus-*TerB* system in human MCF7 cells to artificially generate individual RFBs and investigated the activation of the S-phase checkpoint signaling mechanism. We show that Tus/Ter creates an efficient site-specific RFB in human cells and observed local activation of ATR signaling, which was responsible for the phosphorylation of DNA damage marker γH2AX at the stall sites. When a replication fork pauses at the local Tus-*TerB* block, we do not detect any alteration in global replication profiles. Our system allows us to study the ATR-checkpoint activity as a local response to a single RFB.

## Results

### Tus is found enriched at *TerB* sites integrated in human cells

The 23 bp *TerB* sequence in interaction with the Tus protein have been successfully used in yeast and mice to create artificial RFBs (*Larsen et al., 2014a*; *Willis et al., 2018*; *Willis et al., 2017*; *Willis et al., 2014*; *Willis and Scully, 2016*). To study the effect of site-specific replication blocks at a genomic locus in human cells, we used a previously established MCF7 clone with a unique integrated copy of the pWB15 plasmid carrying two sets of five *TerB* repeat sequences in opposite, non-permissive orientations with respect to the incoming replication forks (*Figure 1A* and *Figure 1—figure supplement 1*). Whole-genome sequencing of the clone was carried out to confirm the single-copy integration of the plasmid at hg38 chr12:95325001. Firstly, we checked whether Tus protein was able to bind *TerB* sequences in this artificial system in human cells. Chromatin immunoprecipitation (ChIP) followed by quantitative polymerase chain reaction (qPCR) was performed using an antibody against the His-tag 24 hr after MCF7 5C-TerB cells were transfected with a Tus-His expression plasmid. The Tus signal is specifically enriched over 20 times compared to the vector control (VC) condition in proximity to the *TerB* sequences (PP9, 2, and 47) (*Figure 1B*).

To corroborate this observation, we performed the proximity ligation assay (PLA) (see 'Methods'). Two tagged-Tus (HA-Tus and Tus-His) or a VC expression plasmid were simultaneously transfected and antibodies against the two tags, HA and His, were used to generate a PLA signal (*Figure 1C and D*). We observed 19% of the cells transfected with the two tagged Tus were harboring one or two PLA signal versus 9% of the cells transfected with the VC (*Figure 1E*), a significant enrichment of the PLA signal when Tus was bound to *TerB*. These results indicate a specific interaction of the Tus protein with the *TerB* sequences in human cells.

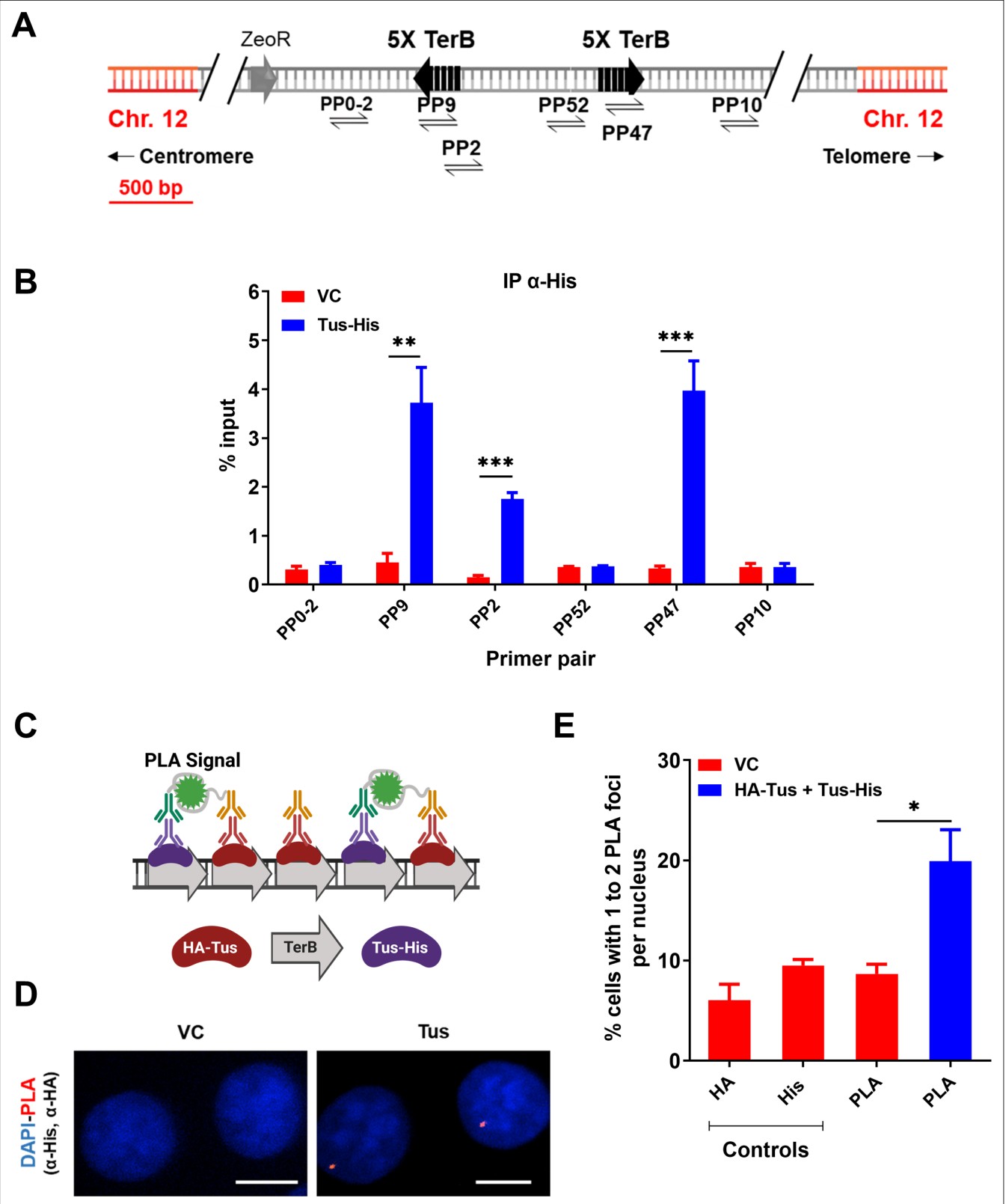

**Figure 1.** Tus is bound to *TerB* sites integrated in human cells. (**A**) Schematic of linearized plasmid (pWB15) integrated into MCF7 cells (MCF7 5C-TerB) with two cassettes containing five tandem *TerB* sequences in the non-permissive orientation (black arrows). Black half-arrowheads depict polymerase chain reaction (PCR) products expected from stated primer pairs used in (**B**). (**B**) Chromatin immunoprecipitation (ChIP) with His antibody on MCF7 5C-TerB cells transfected with either a vector control (VC) or HA-Tus-His expression plasmids. ChIP-qPCR was conducted using the indicated primer

*Figure 1 continued on next page*

*Figure 1 continued*

pairs. Data show the percentage of input (n = 3). (**C**) Schematic of proximity ligation assay (PLA) to visualize Tus bound to *TerB* sites using HA and His antibodies. (**D**) Representative images of the PLA foci across stated conditions. Scale Bars, 10 µm (**E**) Percentage of cells with 1–2 PLA foci per nucleus. (n = 3, ≥150 cells per experiment). Bar graphs represents mean ± standard error of the mean (SEM), Student t-test (see methods).

The online version of this article includes the following source data and figure supplement(s) for figure 1:

**Source data 1.** Tables related to *Figure 1B and E*.

**Figure supplement 1.** Generation of the MCF7 5C-TerB cell line.

## The Tus-*TerB* barrier is an efficient block to incoming replication forks

It has previously been demonstrated that the Tus-*TerB* interaction can induce an RFB in mouse embryonic cells using a chromosomally integrated system (*Willis et al., 2014*). We reasoned that our integrated system in MCF7 cells would be more prone to replication fork arrests near the *TerB* sequences. We used single-molecule analysis of replicated DNA (SMARD) to investigate evidence of replication fork stalling in the genomic DNA (*Norio and Schildkraut, 2001*). The self-labeling SNAP-tag was used to label Tus. After induction of Myc-NLS-TUS-SNAP by using Dox for 3 d, asynchronous MCF7 5C-TerB cells were sequentially pulse-labeled using two nucleoside analogs (iododeoxyuridine [IdU] and chlorodeoxyuridine [CldU]) for 4 hr each (*Figure 2A*). The isolated DNA was digested by restriction endonuclease SfiI, and the 200 kb fragment obtained was stretched on slides for staining. Two DNA FISH probes were used to detect the 200 kb segment of interest containing the *TerB* arrays: one probe of 40 kb within the chromosome 12 (hg38 chr12:95,199,827–95,239,226) and one probe of 7 kb made from pWB15 (*Figure 2B*). The direction of replication forks could be determined using the labeling patterns and its position within the locus of interest using our FISH probes. After inducing the expression of Myc-NLS-TUS-SNAP from an integrated plasmid in MCF7 5C-TerB cells (*Figure 2—figure supplement 1A and B*), we analyzed its impact on the replication fork progression within the region containing *TerB* repeats (*Figure 2C and D*). The yellow arrows indicate the transition of labeling from IdU to CldU incorporation showing the replication fork direction. When Tus was expressed, these transition sites were found accumulated near the *TerB* regions (*Figure 2D*, arrows within the white oval) compared to the random signal spread on the 200 kb region when Tus is not expressed (*Figure 2C*). We quantified the percentage of molecules with replication forks at each 5 kb interval in the 200 kb SfiI segment containing *TerB* sequences (*Figure 2E and F*) and confirmed an accumulation of forks at the *TerB* sequences only when Tus was expressed (*Figure 2F*, black arrow), supporting that the binding of Tus on *TerB* arrays can impair the replisome progression at genomic loci. Interestingly, expression of the Tus protein alone does not alter the direction of replication fork progression along the 200 kb SfiI segment where forks predominantly progress from the 3′ to 5′ direction in cells both with and without Tus expression as analyzed by the percentage of molecules with IdU incorporation at each 5 kb interval (*Figure 2—figure supplement 1C and D*). Furthermore, no significant difference was observed in the global replication fork speed in cells with or without Tus expression (*Supplementary file 1a*). Together, these data indicate that when Tus bound the *TerB* sequences integrated within chromosome 12, it creates a physical barrier and an obstacle to the replication fork progression.

The eukaryotic replisome is a multiprotein apparatus consisting of DNA helicases, DNA polymerases and accessory proteins to duplicate the genome. We hypothesized that an RFB would induce an accumulation of these proteins on chromatin around *TerB* sequences. To test our hypothesis, we performed ChIP using an antibody against mini-chromosome maintenance protein 3 (MCM3), a CMG core-complex protein, in the absence or presence of Tus. When Tus was expressed, MCM3 was threefold more enriched upstream of the *TerB* arrays at the PP0-2 and PP10 sites compared to the surrounding primer pairs (*Figure 2G and H*). Additionally, we find that there is an enrichment of the DNA repair scaffold protein FANCM at the *TerB* sites mirroring the MCM3 enrichment (*Figure 2—figure supplement 2*), indicative of the stalled replication forks as previously described (*Collis et al., 2008*; *Nandi and Whitby, 2012*; *Panday et al., 2021*; *Xue et al., 2015*). Together, these results show that Tus expression induces an RFB at the *TerB* arrays impairing the replisome progression during S-phase.

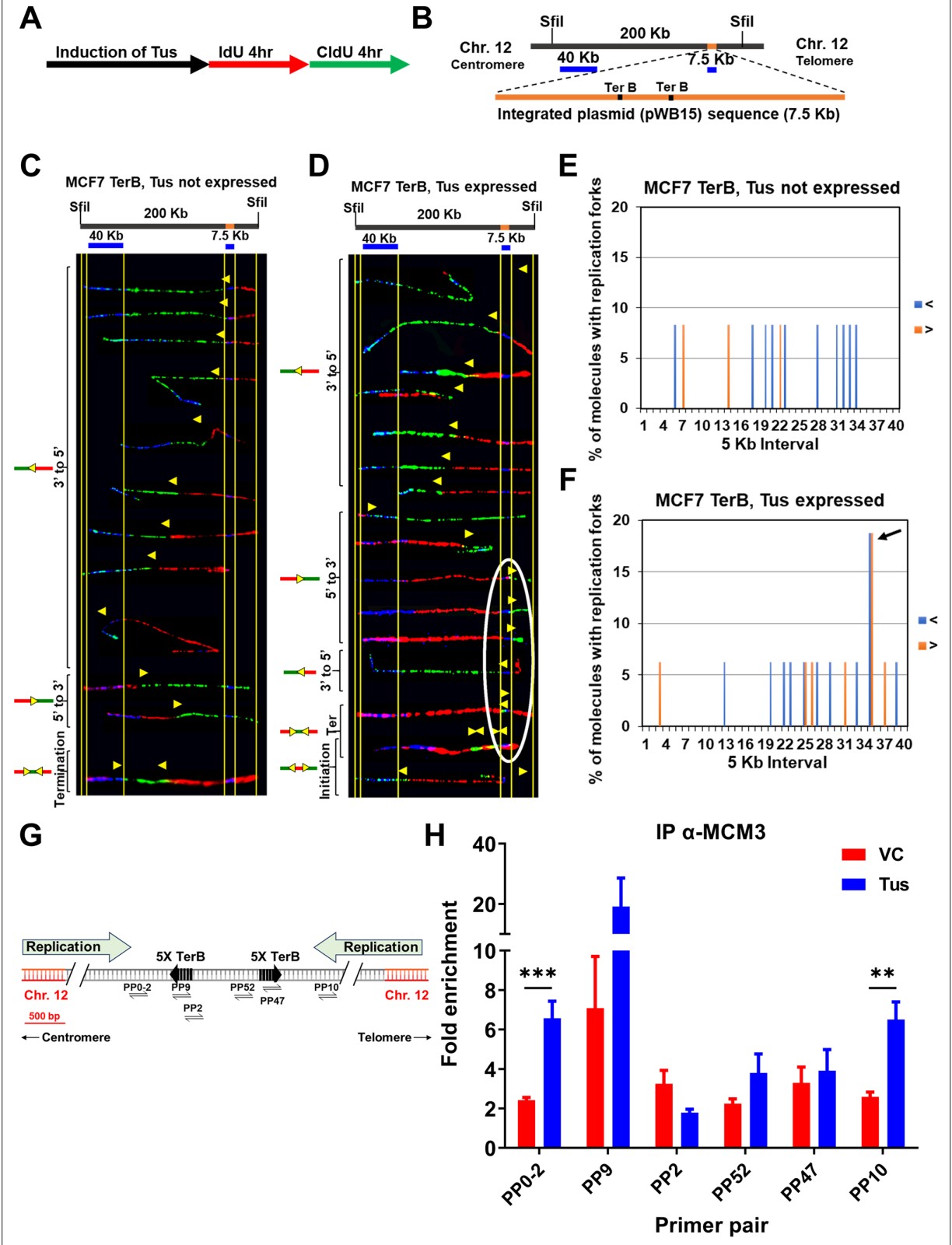

**Figure 2.** Replication fork pauses at the Ter sequence in the presence of the Tus protein. (**A**) Schematic of pulse labeling of MCF7 5C-TerB cells with Tus protein expression for 3 d. (**B**) Locus map of a 200 kb SfiI segment from MCF7 5C-TerB Chromosome 12 with the integrated plasmid (pWB15) DNA (orange). A 40 kb FISH probe made from fosmid WI2-1478M20 and a 7.5 kb FISH probe made from plasmid pWB15 are shown in blue. (**C, D**) Top: locus map of the DNA segment containing *Ter* sequence with the location of the FISH probes. Bottom: photomicrographs of labeled DNA molecules from

*Figure 2 continued on next page*

*Figure 2 continued*

MCF7 5C-TerB, Tus not expressed (**C**) and MCF7 5C-TerB, Tus expressed (**D**). Yellow arrows indicate the position of replication forks at the transition of labeling from IdU to CldU incorporation showing replication fork direction. Molecules are arranged in the following order: forks progressing from 3' to 5', forks progressing from 5' to 3', termination events, and initiation events. Replication forks (yellow arrows) in the white oval are all at the same location (at the *TerB* sequence) on molecules from different cells indicating that replication forks are pausing at this *TerB* sequence. (**E, F**) Percentage of molecules with replication forks at each 5 kb interval in the 200 Kb SfiI segment containing *TerB* sequence, quantified from molecules in MCF7 5C-TerB (**E**) Tus not expressed and (**F**) Tus expressed. Percentage of molecules with replication forks progressing 3' to 5' (< blue) and 5' to 3' (> orange) are shown. In the cells expressing Tus, a high percentage of molecules contain replication forks pausing in both directions in the 5 kb interval that contains Ter sequences (black arrow [**F**], white oval [**D**]). (**G**) Schematic similar to *Figure 1A* with progression of the endogenous origins of replication within the chromosome 12 shown (green arrows). (**H**) Chromatin immunoprecipitation (ChIP) using MCM3 antibody on MCF7 5C-TerB cells transfected with vector control (VC) or HA-Tus-His plasmids. ChIP-qPCR was conducted using the indicated primer pairs. Data shows the fold enrichment relative to the IgG controls (n = 3). Bar graphs represents mean ± standard error of the mean (SEM), Student t-test (see methods).

The online version of this article includes the following source data and figure supplement(s) for figure 2:

Source data 1. Table related to *Figure 2H*.

Figure supplement 1. Expression of Tus in MCF7 5C-TerB cells.

Figure supplement 1—source data 1. Unedited western blot images for *Figure 2—figure supplement 1B*.

Figure supplement 2. Enrichment of FANCM at the Ter sequence in the presence of the Tus protein.

Figure supplement 2—source data 1. Table related *Figure 2—figure supplement 2*.

## γH2AX is enriched at *TerB* sites after Tus expression before fork collapse

One of the earliest responses to replication stress is the phosphorylation of the histone variant H2AX on serine 139 (γH2AX) by members of the phosphoinositide 3-kinase (PI3K)-like family (PIKK) (***Ward and Chen, 2001***). As we demonstrated that *TerB* arrays can block incoming replication forks, we asked whether we could see a γH2AX signal enrichment due to fork arrest. Using a co-immunoprecipitation assay with the chromatin fraction of cross-linked cells expressing either GFP tag or GFP-Tus, we found that GFP-Tus and γH2AX were immunoprecipitated together using GFP or γH2AX antibodies (***Figure 3A***).

To gain insight into the γH2AX signal at the RFB induced by the Tus-*TerB* interaction, cells were transfected with either VC or HA-Tus expression plasmids, and PLA was performed using antibodies against HA and γH2AX (***Figure 3B***). We found that 16% of the HA-Tus transfected cells were harboring at least one PLA signal versus 3% of the VC transfected cells (***Figure 3C and D***).

To better characterize the γH2AX enrichment around *TerB* sites, we performed ChIP-qPCR assays using a γH2AX antibody 24 hr after Tus or VC expression. We observed an enrichment of γH2AX, strictly co-localizing with the Tus-Ter interaction sites (PP9 and PP47), suggesting a local role of γH2AX in response to stalled RFs (***Figure 3E and F***). This is in stark contrast to the observed spreading of the γH2AX signal after a site-directed DSB (***Berkovich et al., 2007***; ***Chailleux et al., 2014***; ***Clouaire et al., 2018***; ***Savic et al., 2009***). To corroborate this observation in our integrated cassette, we generated a site-specific DSB by expressing Cas9 targeted to the integrated cassette (***Figure 3—figure supplement 1A–C***). We assessed γH2AX using the ChIP-qPCR assay 24 hr after transfection and noticed enrichment at the distal primer pairs (PP0-2 and PP10), contrasting with the very tight local signal observed at the Tus-*TerB* RFB (***Figure 3—figure supplement 1D***). However, we did not find any significant γH2AX signal between VC or Tus expression (***Figure 3—figure supplement 2A and B***), supporting the lack of global response. This suggests that the RFB generated by the interaction between Tus and the *TerB* arrays activates a stress response that stimulates the phosphorylation of H2AX, concentrated around the fork block.

## H2AX is phosphorylated in an ATR-dependent manner without a global activation of the intra-S checkpoint

ATR is known to phosphorylate H2AX in response to replication stress and plays a pivotal role in the surveillance of DNA replication (***Ward and Chen, 2001***). To understand whether the global ATR-dependent checkpoint was activated in response to the Tus-*TerB* RFB, we analyzed the levels of phospho-proteins involved in the intra-S checkpoint activation pathway (RPA, CHK1, and ATR) by immunoblotting fractionated cell extracts 24 hr after VC or Tus expression. Cells treated with

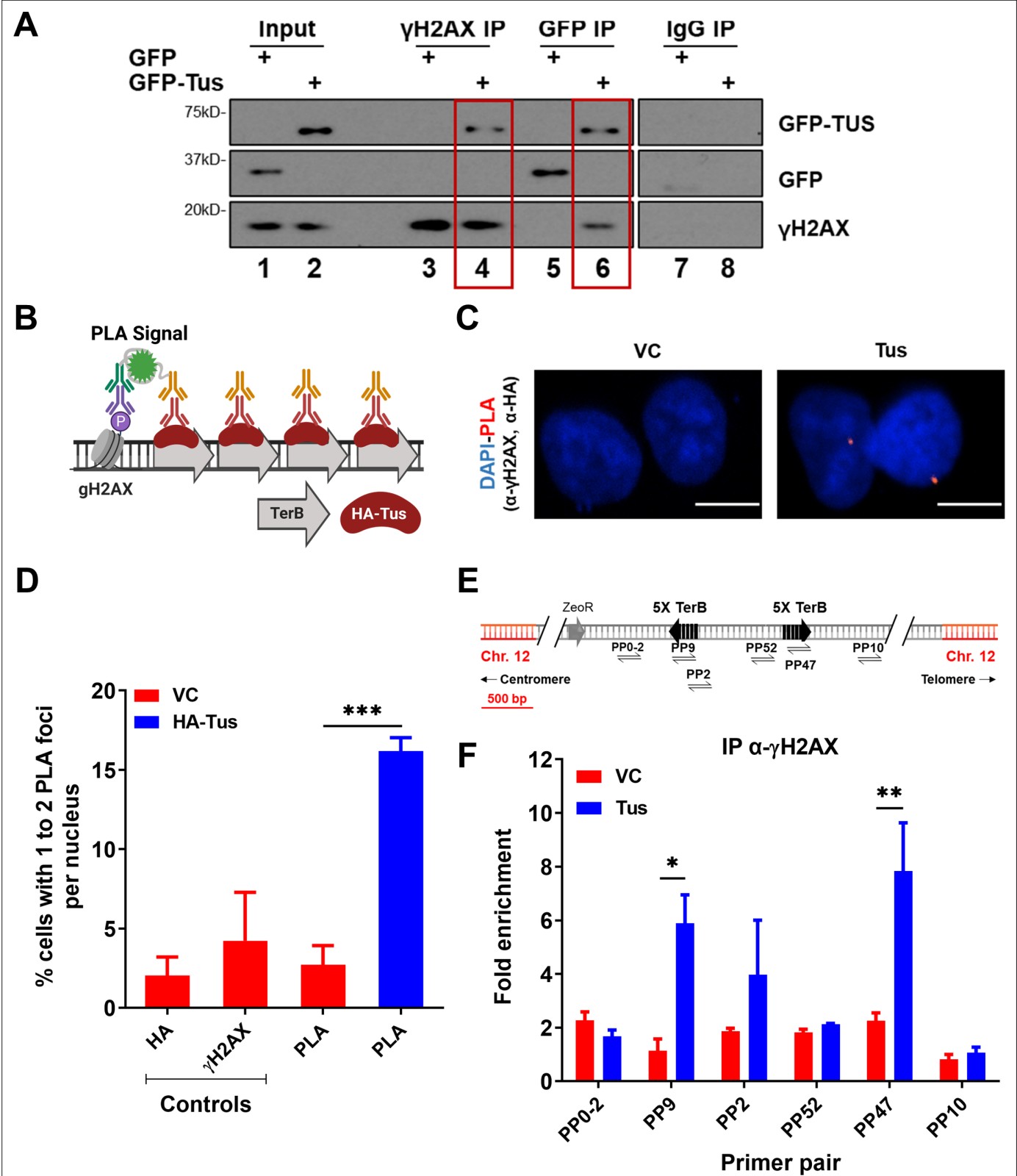

**Figure 3.** γH2AX is enriched at *TerB* sites after Tus expression. (**A**) MCF7 5C-TerB cells transfected with GFP or GFP-Tus, γH2AX and IgG antibodies were used to immuno-precipitate proteins and analyzed by immunoblotting with indicated antibodies. (**B**) Schematic of proximity ligation assay (PLA) to visualize HA-Tus bound to *TerB* in the proximity of γH2AX sites using HA and γH2AX antibodies. (**C**) Representative images of the PLA foci across stated conditions. Scale Bars, 10 μm. (**D**) Percentage of cells with 1–2 PLA foci per nucleus (n = 3, ≥150 cells per experiment). (**E**) Schematic of MCF7 5C-TerB

*Figure 3 continued*

depicting the positions of the primer pairs with respect to the integrated *TerB* locus. (**F**) γH2AX levels along the integrated *TerB* plasmid were analyzed by ChIP-qPCR in MCF7 5C-TerB cells transfected with vector control (VC) or Tus expression plasmids using the indicated primer pairs. Data shows the fold enrichment relative to IgG controls (n = 3). Bar graphs represents mean ± standard error of the mean (SEM), Student t-test (see methods).

The online version of this article includes the following source data and figure supplement(s) for figure 3:

**Source data 1.** Unedited western blot images for *Figure 3A*.

**Source data 2.** Tables related to *Figure 3D and F*.

**Figure supplement 1.** Distinct patterns of γH2AX enrichment at a Cas9-mediated DSB versus the Tus-*TerB* fork barrier.

**Figure supplement 1—source data 1.** Unedited agarose gel for *Figure 1—figure supplement 1C*.

**Figure supplement 1—source data 2.** Table related to *Figure 3—figure supplement 1D*.

**Figure supplement 2.** Genome-wide gH2AX foci were unaffected with Tus expression.

**Figure supplement 2—source data 1.** Table related to *Figure 3—figure supplement 2B*.

2 mM hydroxyurea (HU) for 4 hr demonstrated an increase in phospho-protein levels, acting as a control for checkpoint activation (*Figure 4A*, lanes 3 and 4). Tus expression alone did not activate the global ATR-dependent checkpoint as the level of phospho-protein remained the same as the VC control (*Figure 4A*, lanes 1 and 2). We did not find any observable difference in cell cycle progression between VC or Tus expression alone (*Figure 4—figure supplement 1A and B*), supporting the lack of global response.

We hypothesized that if the local γH2AX enrichment at *TerB* sites was ATR-dependent, a reduction in ATR activity would lead to a decreased γH2AX signal. To validate the ATR inhibitor activity (ATRi), VE-822 (*Fokas et al., 2012*), we performed immunoblotting using total protein extracts of cells treated with 2 mM HU and compared the level of phospho-ATR TH1989 with or without an ATR inhibitor. A 4 hr treatment with ATRi reduced the level of phospho-ATR TH1989, reflecting a decrease in ATR activity (*Figure 4B*). ChIP-qPCR assays were performed using a γH2AX antibody 24 hr after VC or Tus expression, and cells were harvested after a 4 hr ATRi treatment. Consistent with our hypothesis, the γH2AX enrichment in the vicinity of *TerB* sites after Tus expression was remarkably decreased upon ATRi treatment (*Figure 4C–E*, PP9 and PP47), implying that the Tus-*TerB* RFB activates a localized stress response that stimulated the local phosphorylation of H2AX. Together, these data suggest a local activation of the intra-S checkpoint via the ATR kinase and rapid phosphorylation of H2AX, which could mediate the recruitment of repair factors near the damage site.

## Discussion

Replication stress occurs during the S-phase of the cell cycle, when the replication machinery encounters DNA lesions that cause stalling of replicative polymerases and can be a significant cause of genomic instability. If the stalled forks cannot be processed, they can result in DNA breakage, mutations, and chromosomal rearrangements, leading to the development of many different human cancers (*Gaillard et al., 2015*; *Tubbs and Nussenzweig, 2017*). In this study, we used the reconstituted *E. coli*-derived protein-DNA barrier, Tus-*TerB,* to study the replication stress response at a single RFB. We have shown that the Tus protein efficiently binds at the *TerB* sequences using ChIP-qPCR and PLA in MCF7 cells (*Figure 1*). We confirmed that this integrated system was able to cause site-specific RFB at *TerB* sequences when Tus was introduced with the use of the SMARD technique and observed the accumulation of replication fork protein MCM3 upstream of *TerB* sequences (*Figure 2*). Our results indicate that the integrated Tus-*TerB* system acts as an efficient site-specific RFB, providing valuable insights into the local processing of a stalled single fork in human cells and its difference from global replication stress (*Figure 5*).

We show that the site-specific replication stress leads to the accumulation of γH2AX at the site of fork block when Tus was expressed by co-immunoprecipitation of Tus and γH2AX (*Figure 3A*). Additionally, we confirmed that γH2AX is localized at the site of Tus-*TerB* using PLA and ChIP-qPCR assays (*Figure 3D and F*). We suggest that the γH2AX signal is being constrained to a region of less than a kilobase from the *TerB* sites by tortional stress generated when the replication fork encounters the barrier and is not influenced by well-defined topological domains that are one measure of the functional units of the genome (*Dixon et al., 2012*; *Rao et al., 2014*). Furthermore, there was no

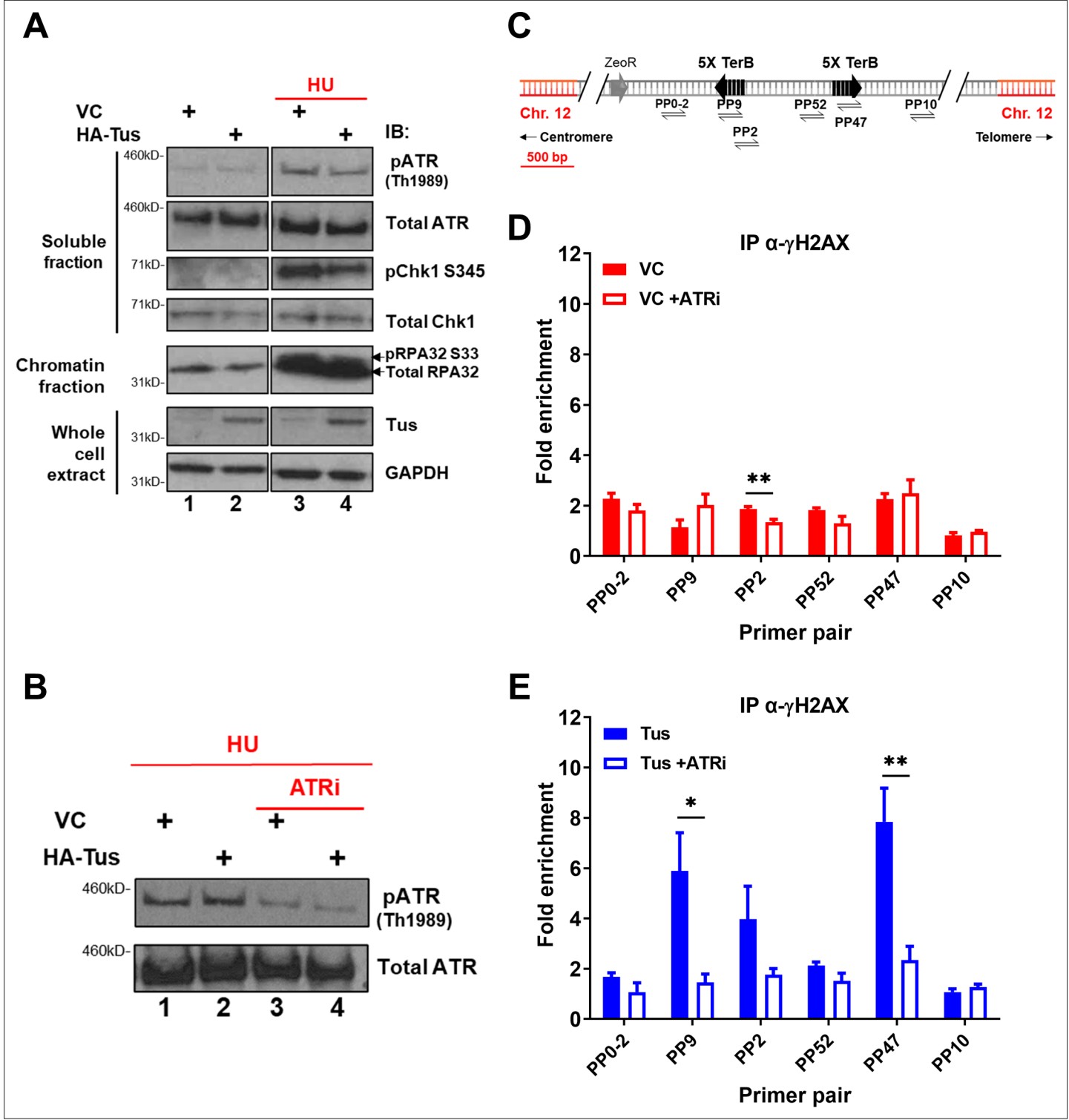

**Figure 4.** γH2AX phosphorylation at Tus-*TerB* is ATR-dependent. (**A**) MCF7 5C-TerB cells transfected with vector control (VC) or HA-Tus were treated for 4 hr with 2 mM hydroxyurea (HU) before lysis and fractionation (whole-cell extract [WCE], soluble fraction, and chromatin fraction). Total and phosphorylated protein levels were examined by immunoblotting as indicated. Tus expression was confirmed in WCE. (**B**) Depletion of pATR Th1989 in MCF7 5C-TerB cells treated with 2 mM HU ± 40 nM of ATR inhibitor was examined with immunoblotting. (**C**) Schematic of MCF7 5C-TerB depicting the positions of the primer pairs with respect to the integrated *TerB* locus. (**D, E**) γH2AX levels were analyzed by ChIP-qPCR in MCF7 5C-TerB cells transfected with (**D**) VC expression plasmid (**E**) Tus expression plasmid, ±4 hr treatment with ATRi with indicated primer pairs. Data shows the fold enrichment over the IgG controls (n = 3). Bar graphs represents mean ± standard error of the mean (SEM), Student t-test (see methods).

*Figure 4 continued on next page*

*Figure 4 continued*

The online version of this article includes the following source data and figure supplement(s) for figure 4:

**Source data 1.** Unedited western blot images for *Figure 4A*.

**Source data 2.** Unedited western blot images for *Figure 4B*.

**Source data 3.** Tables related to *Figure 4D and E*.

**Figure supplement 1.** Cell cycle progression was unaffected by Tus expression.

**Figure supplement 1—source data 1.** Table related to *Figure 4—figure supplement 1B*.

observable difference in global replication profiles or the activation of global checkpoint markers such as pRPA, pCHK1, and pATR, with or without Tus protein expression, contrasting with the differences observed in the presence and absence of HU-induced replication stress (*Figure 4A*, *Supplementary file 1a*, *Figure 3—figure supplement 2*, and *Figure 4—figure supplement 1*). We conclude that Tus-*TerB*-induced replication fork stall does not activate global ATR-dependent S-phase checkpoint signaling, suggesting that a local response is occurring at the stalled replication fork. Interestingly, upon ATR inhibition, we observed that the enrichment of phosphorylated H2AX at *TerB* sites was significantly reduced when Tus was expressed, which indicates local ATR signaling is responsible for

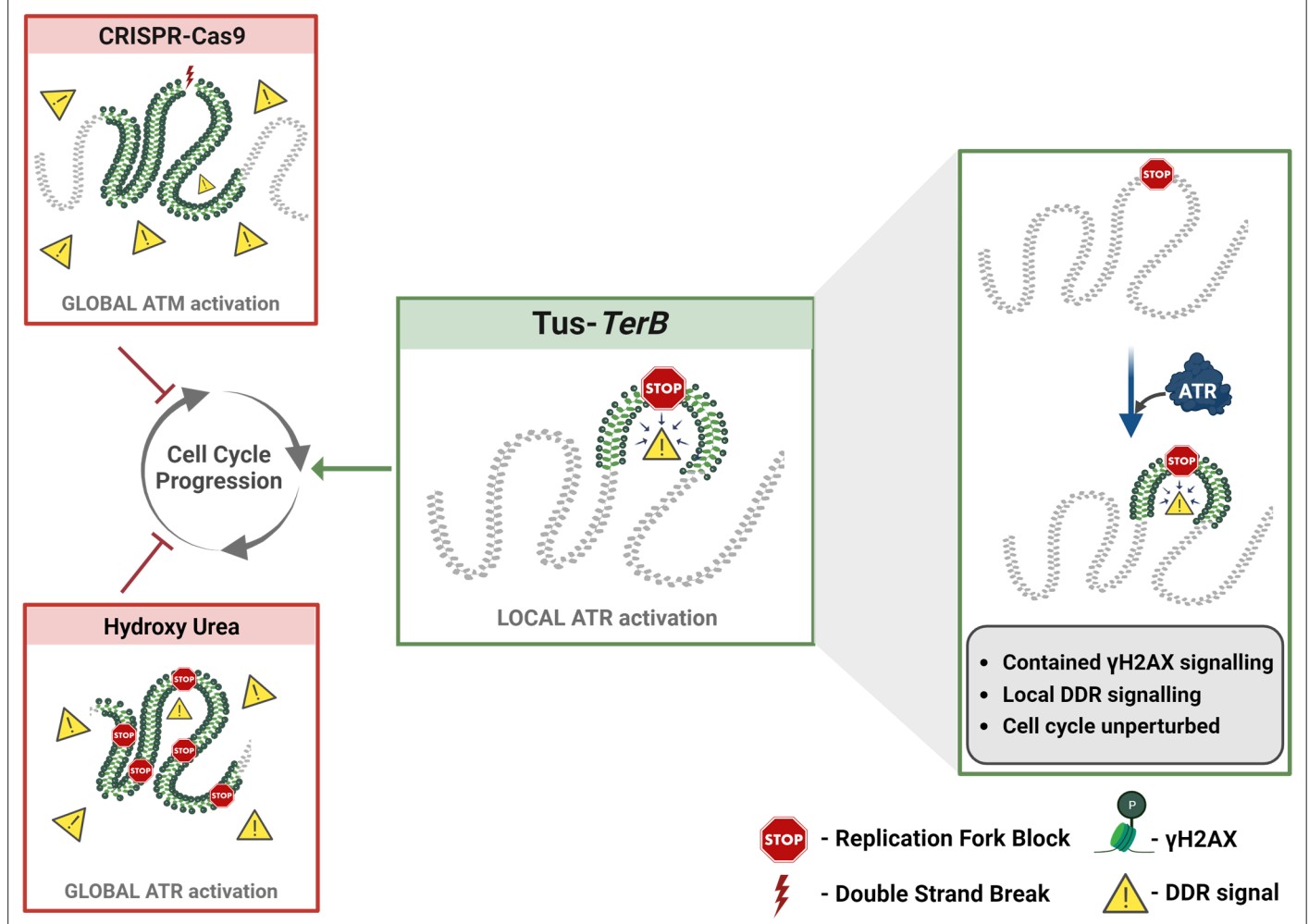

**Figure 5.** A local ATR-dependent checkpoint is activated by the Tus-*TerB* replication fork barrier (RFB). A model depicting the cellular response to a site-specific DSB using CRISPR-Cas9 and a global replication stress with hydroxyurea, both of which lead to a DNA damage response (DDR, yellow triangles), increased gH2AX (green) levels globally in the cell and, if left unresolved, can alter cell cycle progression. In contrast, the site-specific replication fork block, Tus/*TerB*, elicits a local ATR-dependent DDR, which is responsible for the phosphorylation and accumulation of γH2AX at the stalled site. This local signaling does not affect the progression of the cell cycle and is not altered during the local checkpoint response.

the phosphorylation of H2AX at the stalled site (*Figure 4C and E*). A similar localized response had previously been observed in yeast, with a site-specific DSB in G1 cells, where there was no activation of the global checkpoint signaling, but γH2AX signaling was observed (*Janke et al., 2010*). This localized ATR-dependent γH2AX differs from the well-observed ATM-dependent spreading of γH2AX signal after a site-directed double-strand break (*Berkovich et al., 2007*; *Chailleux et al., 2014*; *Clou-aire et al., 2018*; *Savic et al., 2009*) and suggests a distinct local intra-S phase checkpoint is being activated in response to the single RFB.

There are additional mechanisms to be understood in this newly described local replication stress response, which includes whether the activation of ATR is dependent on ATRIP bound to ssDNA (*Ball et al., 2005*; *Zou and Elledge, 2003*). In the Tus-*TerB* system, there is no known dissociation of the replicative helicase from the polymerase, which is responsible for the ssDNA signal. Therefore, just like ATM can be activated independently of the MRE11 complex (*Bakkenist and Kastan, 2003*), we expect that ATR is activated independently of ATRIP, likely due to local changes in chromatin that will be investigated in future studies.

During extensive DNA damage, the genome-wide checkpoint response activates ATR/CHK1 globally, which leads to the slowing of all replication forks in the cell, inhibition of cell cycle progression, and suppression of late origin firing; to provide sufficient time for DNA repair. It has been proposed that a local checkpoint response occurs when one or a few replication forks encounter a DNA lesion and activates ATR/CHK1 signaling at local sites of fork stalling. The local ATR response has been hypothesized to be transient, in which the fork moves slowly only at stalled sites to promote fork stabilization, restart the stalled fork, and suppress recombination without triggering the global checkpoint response (*Iyer and Rhind, 2017*; *Iyer and Rhind, 2013*; *Kaufmann et al., 1980*; *Merrick et al., 2004*; *Saxena and Zou, 2022*; *Willis and Rhind, 2009*; *Zeman and Cimprich, 2014*).

We predict that the local ATR-dependent checkpoint signaling can result in a rapid and controlled response to allow RFB resolution through the recruitment of repair factors to the damaged site. The factors might include those involved in fork stabilization, fork reversal, fork cleavage, and homology-dependent replication restart (*Saxena and Zou, 2022*). The Tus-*TerB* system has previously been employed in mouse embryonic stem cells to measure and investigate the DNA repair pathway choice (*Chandramouly et al., 2013*; *Panday et al., 2021*; *Willis et al., 2017*; *Willis et al., 2014*). This included both error-free and error-prone homologous recombination induced by a mammalian chromosomal RFB, as well as identifying the role of the structure-specific endonuclease complex SLX4-XPF and FANCM in the repair of microhomology-mediated tandem duplications that occur at replication arrest (*Elango et al., 2022*; *Nandi and Whitby, 2012*; *Willis et al., 2017*). The Tus-*TerB* block is robust and is not dissociated by the action of DNA helicases such as PIF1, corroborated in previous yeast studies that have demonstrated that the 'sweepase' RRM3 is unable to resolve this RFB (*Larsen et al., 2014b*). Therefore, it is likely that the RFB is resolved either by incoming replication forks leading to termination at the *TerB* site or by fork cleavage (*Larsen et al., 2014b*; *Willis et al., 2017*; *Willis et al., 2014*). Fork reversal at the Tus-*TerB* site has been observed in the yeast system (*Larsen et al., 2014b*; *Marie and Symington, 2022*), but not for the mammalian systems, and it is unlikely that shifting the position of the RFB would make it easier to resolve. Therefore, fork cleavage is the most likely mechanism for resolving the RFB.

The upstream pathway of activation of the DNA damage response, as a result of a single RFB, is not completely understood. Previous work has shown that FANCM acts as a scaffolding protein for recruitment of different repair protein complexes involved in the stalled replication fork rescue (*Panday et al., 2021*; *Willis et al., 2017*). We observed that the FANCM protein is enriched at the site of the RFB induced by Tus-*TerB* (*Figure 2—figure supplement 2*), confirming a similar phenomenon observed in mouse cells. However, it is yet to be determined whether FANCM recruitment to the RFB precedes local H2AX phosphorylation by ATR or whether FANCM is recruited as a consequence of the γH2AX signal to help elicit the downstream DNA damage response (DDR). We also anticipate that the protein complex, 9-1-1 (RAD9A, HUS1, RAD1), and the subsequently recruited TOPBP1, may also play a role at this local ATR-dependent intra-S checkpoint (*Helt et al., 2005*; *Parrilla-Castellar et al., 2004*).

While the Tus-*TerB* system is an important tool in deciphering signaling at individual replication forks, it will be important to test whether the local-S phase model remains supported when exploring endogenous lesions that occur when replication forks encounter secondary structures such as R-loops,

G-quadruplexes, and common fragile sites characterized by trinucleotide repeat expansion [(CAG)$_n$/(CTG)$_n$] (**Brickner et al., 2022**; **Bryan, 2019**; **Kim et al., 2017**). Here the mechanisms surrounding the identification and resolution of the RFB may differ. Overall, our results indicate that Tus-*TerB* system acts as an efficient RFB and activates local ATR checkpoint signaling at the stall site, leading to phosphorylation and accumulation of the DNA damage sensor protein γH2AX, which is dependent on the ATR kinase. The local γH2AX accumulation at the stalled region would lead to the recruitment of DNA repair factors for resolution of the fork (**Figure 5**). Together, our findings reveal the signaling mechanism of the Tus/*TerB*-induced replication block and showed that the site-specific fork block is dependent on the local ATR S-phase checkpoint signaling.

## Methods

### Cell culture and transfection conditions

MCF7 5C-TerB cells were grown at 37°C and 5% $CO_2$ in complete DMEM high glucose supplemented with 10% FBS, 2 mM GlutaMAX, 20 mM HEPES, 100 I.U./ml penicillin, and 100 μg/ml streptomycin supplemented with 400 μg/ml zeocin. The cell line was characterized using the Cell Line Authentication by STR analysis at the Integrated Genomics Operation, MSKCC. Cells were routinely tested for mycoplasma contamination using the Universal Mycoplasma Detection Kit (ATCC 30-1012K).

For PLA, cells were seeded at $1 \times 10^5$ in a 12-well cell culture plate. For T7 assay, cells were seeded at $2.5 \times 10^5$ in a 6-well cell culture plate. For ChIP, IP, and cellular fractionation, cells were seeded at $2.5 \times 10^5$ in a 10 cm cell culture dish.

MCF7 5C-TerB cells were seeded and transfected the day after with 10 μg of pCMV3xnls Tus or pCMV3xnls using the Mirus Bio TransIT-LT1 reagents for 24 hr more. For the generation of DSB, cells were transfected 6 hr prior to harvesting with 120 pmol of the sgRNA TerB1 and 120 pmol of the Cas9 protein (obtained from QB3 Macrolab, Berkeley) using the Lipofectamine CRISPRMAX Cas9 transfection reagents.

### Generation of stable cell lines

PvuI linearized pWB15 was transfected into MCF7 (ATCC HTB-22) cells to generate the stably integrated clone, MCF7 5C-TerB (**Figure 1—figure supplement 1**). Then, 800 μg/ml of zeocin was used to select positively integrated cells and monoclonal colonies were isolated and subsequently maintained in 400 μg/ml zeocin. Single-integrant validation was performed by whole-genome sequencing of the clone.

Stable MCF7 5C-TerB cells-inducible-expressing Myc-NLS-TUS-SNAP were generated by lentiviral transduction. Cells were infected with lentiviral particles containing pInducer10L or pInd Tus-SNAP, and single-cell clonal colonies were selected in complete DMEM containing 1 μg/ml puromycin and 400 μg/ml zeocin. Myc-NLS-TUS-SNAP protein was expressed in stable lines by induction with 1 μg/ml doxycycline for 3 d.

### Plasmid construction

The doxycycline-inducible lentiviral plasmid used to express N-terminally myc-tagged, C-terminally SNAP-tagged nuclear localized, human codon-optimized wild-type Tus (Myc-NLS-TUS-SNAP) was generated as follows. The N-terminally myc epitope-tagged, nuclear-localized, codon-optimized wild-type Tus cDNA sequence was PCR-amplified from pcDNA3-β-MYC-NLS-Tus using primers forward 5′-AGTCGGTACCGAATTCGCCACCATGGAACAAAAGCTG-3′ and reverse 5′-AGTCGGCGGCCGCGCCGCTACCGTCAGCCACGTACAGGTGCA and the SNAP-tag cDNA sequence PCR amplified using primers forward 5′-AGTCGCGGCCGCCGGCCACATGGACAAAGACTGCGAAATGAAGC-3′ and reverse 5′-ACTGCTCGAGTCAACCCAGCCCAGGCTTGC-3′. The Tus and SNAP PCR-amplified fragments were digested with KpnI and NotI, and NotI and XhoI, respectively, and inserted into the inducible expression lentiviral vector pInducer10L (**Drosopoulos et al., 2020**) directly downstream of the doxycycline-inducible promoter, to generate pInd Tus-SNAP. This construct was sequenced to confirm that unintended mutations were not introduced during PCR and cloning.

## Immunoblotting

Cells were harvested by trypsinization, suspended in complete DMEM, washed with PBS, then pelleted and flash frozen in liquid $N_2$ and stored at –80°C. For SDS-PAGE, pellets were thawed on ice and lysed by resuspending in Laemmli buffer (60 mM Tris-HCl pH 6.8, 400 mM 2-mercaptoethanol, 2% SDS, 10% glycerol, 0.01% bromophenol blue) to a final concentration of $10^6$ cells/ml. Lysates were denatured (5 min at 100°C) and passed through a 25-gauge needle (5×) then spun for 2 min at full speed in a microfuge. Aliquots of lysate corresponding to $10^5$ cells were resolved on 4–15% gradient SDS-PAGE gels, proteins transferred to nitrocellulose membrane and blocked in PBS with 5% Blotting-grade Blocker and 0.1% Tween 20. Membranes were then incubated with primary antibodies diluted in PBS with 5% Blotting-grade Blocker. Primary antibodies used were anti-Myc tag and anti-actin. Following incubation with primary antibodies, membranes were washed with PBS + 0.1% Tween 20. Membranes were then incubated with fluorescently labeled goat anti-mouse IRDye 680LT and goat anti-rabbit IRDye800CW secondary antibodies and then washed in PBS + 0.1% Tween 20. Immunoblots were imaged on an Odyssey Lc Infrared scanner.

## SMARD assay

SMARD was performed essentially as previously described (*Drosopoulos et al., 2012*; *Norio and Schildkraut, 2001*; *Twayana et al., 2021*). Exponentially growing cells in complete DMEM with 1 μg/ml doxycycline were sequentially pulse-labeled with IdU (4 hr), followed by CldU (4 hr) each to a final concentration of 30 μM. Following pulsing, cells were suspended in PBS at a concentration of $3 \times 10^7$ cells per ml. An equal volume of 1% molten TopVision low-melting point agarose in PBS was added. The resulting cell suspension in 0.5% TopVision agarose was poured into wells of a chilled plastic mold to make plugs of size 0.5 cm × 0.2 cm × 0.9 cm, each containing $10^6$ cells. Cells in the plug were lysed at 50°C in a buffer containing 1% n-lauroylsarcosine, 0.5 M EDTA, and 0.2 mg/ml proteinase K. Plugs were rinsed in TE and washed with 200 μM phenylmethanesulfonyl fluoride (PMSF). Genomic DNA in the plugs was digested with *SfiI* overnight at 37°C. Digested genomic DNA was cast into 0.7% SeaPlaque GTG agarose gel, and DNA was separated by pulsed field gel electrophoresis (PFGE) using a CHEF-DRII system (Bio-Rad). The 200 kb DNA segment containing *TerB* sequences within the gel was located by performing Southern hybridization on a portion of the gel using a specific probe made from the pWB15 plasmid. This identified the pulse field gel slice containing *TerB* sequences, which was excised and melted (20 min at about 72°C). The DNA in the gel solution was stretched on microscope slides coated with 3-aminopropyltriethoxysilane, denatured with sodium hydroxide in ethanol, fixed with glutaraldehyde, and hybridized overnight with biotinylated DNA FISH probes at 37°C in a humidified chamber. Biotinylated DNA FISH probes were made from the fosmid WI2-1478M20 and from the plasmid pWB15. Following hybridization, slides were blocked with 3% BSA for at least 20 min and incubated with the avidin Alexa Fluor 350. This was followed by two rounds of incubation with the biotinylated anti-avidin D for 20 min, followed by the avidin Alexa Fluor 350 for 20 min. Slides were then incubated with an antibody specific for IdU; a mouse anti-BrdU antibody, an antibody specific for CldU; a rat monoclonal anti-BrdU antibody, and the biotinylated anti-avidin D for 1 hr. This was followed by incubation with secondary antibodies: Alexa Fluor 568 goat anti-mouse IgG (H+L), Alexa Fluor 488 goat anti-rat IgG (H+L), and the avidin Alexa Fluor 350 for 1 hr. Slides were rinsed in PBS with 0.03% IGEPAL CA-630 after each incubation. After the last rinse in PBS/CA-630, coverslips were mounted on the slides with ProLong Gold Antifade reagent. A Zeiss fluorescent microscope and IP Lab software (Scanalytics) were used to capture images of IdU/CldU incorporated DNA molecules. Images were processed using Adobe Photoshop and aligned (using Adobe Illustrator) based on the positions of FISH signals that identify the 200 kb segments that contain the ter sequences.

## Proximity ligation assay (PLA)

MCF7 5C-TerB cells were seeded on poly-L-lysine-coated coverslips and transfected the day after as described before. Then, 24 hr after transfection, cells were washed with PBS, and fixed and permeabilized with 4% paraformaldehyde containing 0.2% Triton X-100 for 20 min on ice and blocked with PBS-BSA 3% overnight at 4°C. Coverslips were incubated with primary antibodies for 1 hr at room temperature (RT). Proximity ligation was performed using the Duolink In Situ Red Starter Kit Mouse/Rabbit (Sigma-Aldrich) according to the manufacturer's protocol. The oligonucleotides and

antibody-nucleic acid conjugates used were those provided in the Sigma-Aldrich PLA kit. Images were quantified by counting the number of foci per nucleus using Nikon software.

## Immunoprecipitation

For immunoprecipitation, MCF7 5C-TerB cells were pre-extracted using CSK100 buffer (100 mM NaCl, 300 mM sucrose, 3 mM MgCl$_2$, 10 mM PIPES pH 6.8, 1 mM EGTA, 0.2% Triton X-100, anti-protease and anti-phosphatase) for 5 min on ice, fixed in 1% formaldehyde for 10 min on ice, and a 1% glycine solution was used to stop the reaction. After scraping the cells in ice-cold PBS and centrifugation, pellets were lysed in SDS lysis buffer (50 mM Tris-HCl pH 7.5, 150 mM NaCl, 0.1% SDS, anti-proteases, and anti-phosphatases) for 10 min on ice and sheared for 3 min. Samples were cleared by centrifugation for 5 min at 4°C. Immunoprecipitations were performed on 10-fold diluted lysates in dilution buffer (50 mM Tris-HCl pH 7.5, 150 mM NaCl, 5 mM EDTA, 0.2% Triton X-100, anti-proteases and anti-phosphatases) with antibodies against GFP, γH2AX, or IgG overnight at 4°C on a wheel. Beads were extensively washed in the dilution buffer and denatured in 2× Laemmli buffer.

Proteins were separated on 4–12% acrylamide SDS-PAGE, transferred on nitrocellulose membrane, and detected with the indicated antibodies described in *Supplementary file 1b* and ECL reagents.

## Cellular fractionation

For the cellular fractionation, MCF7 5C-TerB cells were scraped in PBS, divided into two different tubes (1/3 of the volume for the whole-cell extract [WCE] and the remaining 2/3 for the fractionation) and centrifuged to keep the pellets.

For the WCE, cells were lysed in 1 volume of lysis buffer (50 mM Tris pH 7.5, 20 mM NaCl, 1 mM MgCl$_2$, 0.1% SDS, anti-protease and anti-phosphatase) for 10 min at RT on a wheel and denatured in 2× Laemmli buffer.

For the fractionation, MCF7 5C-TerB cells were pre-extracted in 2 volumes of CSK100 (100 mM NaCl, 300 mM sucrose, 3 mM MgCl$_2$, 10 mM PIPES pH 6.8, 1 mM EGTA, 0.2% Triton X-100, anti-protease and anti-phosphatase) for 15 min on ice, and centrifuged. The supernatant (SN) representing the soluble fraction was kept in a new tube (soluble fraction). The pellet was washed with CSK50 (50 mM NaCl, 300 mM sucrose, 3 mM MgCl$_2$, 10 mM PIPES pH 6.8, 1 mM EGTA, 0.2% Triton X-100, anti-protease and anti-phosphatase) and resuspended in 2 volumes of CSK50 containing benzonase for 10 min on a rotating wheel, and the SN was kept after centrifugation (chromatin fraction). All the fractions were denatured in 2× Laemmli buffer.

Proteins were separated on 4–12% acrylamide SDS-PAGE, transferred on nitrocellulose membrane, and detected with the indicated antibodies described in *Supplementary file 1b* and ECL reagents.

## Chromatin immunoprecipitation assay

ChIPs were performed using the ChIP-IT express according to the manufacturer's instructions. Cells were crosslinked, washed, harvested by scraping, and lysed according to the kit's instructions. Chromatin was sonicated to 200–1500 bp fragments and DNA concentration determined. Equal amount up to 15 ug of DNA was then used for each immunoprecipitation. The antibodies used are listed in *Supplementary file 1b*. DNA fragments were eluted, purified, and analyzed by SYBR Green real-time PCR. The sequences of primers used for qPCR are given in *Supplementary file 1b*. Experiments were repeated at least three times, and each real-time qPCR reaction was performed in duplicate.

## Detection of DSBs using the T7 endonuclease assay

MCF7 5C-TerB cells were seeded and transfected the day after with 120 pmol of Cas9 protein and 120 pmol of the sgRNA TerB1 using the Lipofectamine CRISPRMAX Cas9 transfection reagents (previously described). Then, 24 hr post-transfection, cells were lysed using a lysis buffer (100 mM NaCl, 10 mM Tris-HCl pH 8, 25 mM EDTA pH 8, 0,5% SDS) and 50 ug of Proteinase K overnight at 50°C with shaking. After addition of NaCl and mix for 1 min, the supernatant was put in a new tube and an ethanol precipitation was performed. A PCR to amplify the 900 bp region surrounding the sgRNA TerB1 site was conducted using PP9F and R. Heteroduplex were formed by heating and cooling down the samples, and a T7 endonuclease assay digestion was performed prior to electrophoresis for detection.

## Cell cycle progression

MCF7 5C-TerB cells were transfected as described before. Control cells were treated with 2 mM HU for 4 hr before harvesting. Cells were washed with PBS and fixed in 1 ml cold 70% ethanol for 30 min. Cells were pelleted and washed with PBS. Then, 100 µg/ml RNase A was added and 50 µg/ml PI solution was added directly to the pellet, mixed well, and incubated for 30 min at RT in the dark. A total of 50,000 cells per condition were analyzed by flow cytometry.

## Immunofluorescence (γH2AX)

Cells were fixed with 4% PFA/PBS for 20 min, permeabilized with 0.5% Triton-X 100 for 10 min, washed three times in 1× PBS, and blocked in 10% goat serum overnight at 4°C. The primary antibody (1:500) was incubated for 2 hr at RT. Cells were then washed three times in 1× PBS. The secondary antibody (1:1500) was incubated for 1 hr at RT in dark. Cells were then washed again three times in 1× PBS. Stained cells were mounted with mounting medium containing DAPI. Slides were imaged at 60× (immersion oil) using Nikon A1 spinning disk confocal microscope.

## Image analysis

For PLA and gH2AX experiments, slides were imaged at 60× (immersion oil) with Nikon spinning disk confocal microscope. PLA foci per nucleus and gH2AX foci per nucleus were calculated using Nikon Elements AR Analysis Explorer (version 5.21.03), where DAPI was used as a mask for the nucleus. The number of PLA foci per nucleus was quantified to get the percentage of cells with one or two foci indicating a positive signal at the site-specific replication block. The number of gH2AX foci was counted for each DAPI to obtain the average number of gH2AX foci in each condition.

## NGS sequencing of the MCF7 5C-TerB clone

Genomic DNA was extracted from the cells and submitted to Novogene for sequencing. The samples were processed and whole-genome sequencing was performed using their hWGS service pipeline.

## Data analysis and Statistics

Statistical analysis for the various experiments was performed using GraphPad Prism version 9.4.0. Results are presented as mean ± standard error of the mean (SEM). A p value of <0.05 by Student's *t*-test was considered statistically significant. ns, nonsignificant, *p<0.05, **p<0.005, ***p<0.001.

# Acknowledgements

We are indebted to Ino de Bruijn, Jorge S Reis-Filho, and Yingjie Zhu for help with the genomics data. We thank the members of the Powell and Schildkraut lab for their comments on the manuscript. This work was supported in part by NIH grants R01-CA187069 and P50-CA247749 (to SNP); R01-GM045751 and R01-CA085344 (to CLS); the National Cancer Institute Cancer Center Support Grants, P30-CA008748 at MSK, and P30-CA013330 for use of a core facility at Albert Einstein COM. ST was supported by NIH Training Grant T-32 NIH T32AG023475. Figures were created using BioRender.com

# Additional information

## Funding

| Funder | Grant reference number | Author |
| --- | --- | --- |
| National Institutes of Health | R01-CA187069 | Simon N Powell |
| National Institutes of Health | P50-CA247749 | Simon N Powell |
| National Institutes of Health | R01-GM045751 | Carl L Schildkraut |

| Funder | Grant reference number | Author |
|---|---|---|
| National Institutes of Health | R01-CA085344 | Carl L Schildkraut |
| National Cancer Institute | P30-CA008748 | Simon N Powell |
| National Cancer Institute | P30-CA013330 | Carl L Schildkraut |
| National Institutes of Health | T32AG023475 | Shyam Twayana |

The funders had no role in study design, data collection and interpretation, or the decision to submit the work for publication.

## Author contributions

Sana Ahmed-Seghir, Conceptualization, Data curation, Formal analysis, Validation, Investigation, Visualization, Methodology, Writing - original draft; Manisha Jalan, Conceptualization, Formal analysis, Validation, Visualization, Methodology, Writing - original draft, Writing - review and editing; Helen E Grimsley, Formal analysis, Methodology, Writing - original draft; Aman Sharma, Formal analysis, Methodology, Writing - original draft, Writing - review and editing; Shyam Twayana, Settapong T Kosiyatrakul, Formal analysis, Methodology; Christopher Thompson, Methodology; Carl L Schildkraut, Formal analysis, Supervision, Funding acquisition, Methodology, Project administration, Writing - review and editing; Simon N Powell, Conceptualization, Resources, Supervision, Funding acquisition, Investigation, Writing - original draft, Project administration, Writing - review and editing

## Author ORCIDs

Manisha Jalan ⓘ http://orcid.org/0000-0002-4467-4934
Helen E Grimsley ⓘ http://orcid.org/0000-0003-3485-0305
Aman Sharma ⓘ http://orcid.org/0000-0003-1017-8307
Simon N Powell ⓘ http://orcid.org/0000-0002-8183-4765

Joint Public Review: https://doi.org/10.7554/eLife.87357.3.sa1
Author Response https://doi.org/10.7554/eLife.87357.3.sa2

# Additional files

## Supplementary files

• Supplementary file 1. Supplementary information for *Figure 2* and extended methods. (**a**) Demonstration that the Tus-Ter replication fork block does not activate significant replication elsewhere in the genome. Replication characteristics of 200 kb global DNA segments that represent the total genome. These measurements do not include the segments containing the *TerB* sequence. The table compares MCF7 cells containing the *TerB* sequence with and without Tus induced. This was determined on stretched DNA molecules that had completely incorporated IdU, CldU, or a combination of both nucleotide analogs. (**b**) List of antibodies, oligos, and plasmids used in the study.

• MDAR checklist

## Data availability

The list of antibodies, plasmids and oligonucleotides used in this study are provided in Supplementary file 1b. WGS sequencing files generated in this study have been deposited at the NCBI BioProject database under accession number PRJNA868342 and are publicly available as of the date of publication. Any additional information required to reanalyze the data reported in this paper is available from the lead contact upon request.

The following dataset was generated:

| Author(s) | Year | Dataset title | Dataset URL | Database and Identifier |
|---|---|---|---|---|
| Sana AS, Manisha J, Helen EG, Aman S, Shyam T, Settapong TK, Christopher T, Carl LS, Simon NP | 2023 | Whole Genome Sequencing of the MCF7 5C TerB clone | https://www.ncbi.nlm.nih.gov/bioproject/PRJNA868342 | NCBI BioProject, PRJNA868342 |

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
