## [Editor Report · eLife assessment]

This manuscript reports **important** data on the cellular response to a single site-specific replication fork block in human MCF7 cells. **Compelling** evidence shows the efficacy of the bacterial Tus-Ter system to stall replication forks in human cells. Fork stalling led to lasting ATR-dependent phosphorylation of histone H2AX but not of ATR itself and its downstream targets RPA and CHK1.

---

## [Referee Report · Joint Public Review]

The authors report the first use of the bacterial Tus-Ter replication block system in human cells. A single plasmid containing two divergently oriented five-fold TerB repeats was integrated on chromosome 12 of MCF7 cells. ChIP and PLA experiments convincingly demonstrate the occupancy of Tus at the Ter sites in cells. Using an elegant Single Molecule Analysis of Replicated DNA (SMARD) assay, compelling data demonstrate the replication block at Ter sites dependent on the presence of the protein. As an orthogonal method to demonstrate fork stalling, ChIP data show the accumulation of the replicative helicase component MCM3 and the repair protein FANCM around the Ter sites. Previous published work from the Scully and Hickson laboratories showed that Ter sites do not perturb replication fork progression and consistently the data show that the observed effects are dependent on expression of the Tus protein. The SMARD data reveal that about one third of the forks are arrested at Tus/Ter but it is unclear for how long forks remain stalled. Fork stalling led to a highly localized gammaH2AX response, as monitored by ChIP using primer pairs spread along the integrated plasmid carrying the Ter sites. This response was shown to be dependent on ATR using the ATR inhibitor VE-822. This contrasts with a single Cas9-induced DSB between the two Ter sites, which causes a more spread gammaH2AX response measured at two sites flanking the DSB. The difference between the DSB and the Tus-induced stall is very significant. Interestingly, despite evidence for ATR activation through the gammaH2AX response, no evidence for phosphorylation of ATR-T1989, CHK1-S345, or RPA2-S33 could be found under fork stalling conditions. The global replication inhibitor hydroxyurea (HU) elicited phosphorylation of ATR-T1989, CHK1-S345, or RPA2-S33. In this context, it would have been of interest to examine if a single DSB in the Ter region leads to phosphorylation of ATR-T1989, CHK1-S345, or RPA2-S33 and cell cycle arrest. The replication inhibitor HU led to an increase in gamma H2AX foci consistent with a global replication stress response. Overall, this is a well written manuscript, and the data provide convincing evidence that the Tus-Ter system poses a site-specific replication fork block in MCF7 cells leading to a localized ATR-dependent DNA damage checkpoint response that is distinct from the more global response to HU or DSBs.

---

## [Author Response]

The following is the authors’ response to the original reviews.

**Public Review:**
The authors report the first use of the bacterial Tus-Ter replication block system in human cells. A single plasmid containing two divergently oriented five-fold TerB repeats was integrated on chromosome 12 of MCF7 cells. ChIP and PLA experiments convincingly demonstrate the occupancy of Tus at the Ter sites in cells. Using an elegant Single Molecule Analysis of Replicated DNA (SMARD) assay, convincing data demonstrate the replication block at Ter sites dependent on the presence of the protein. As an orthogonal method to demonstrate fork stalling, ChIP data show the accumulation of the replicative helicase component MCM3 and the repair protein FANCM around the Ter sites. It is unclear whether the Ter sites integrated by a single copy plasmid have any effect on the replication of this region but the data show that the observed effects are dependent on expression of the Tus protein. The SMARD data do not reveal what proportion of forks are arrested at Tus/Ter, or how long the fork delay is imposed. Fork stalling led to a highly localized gammaH2AX response, as monitored by ChIP using primer pairs spread along the integrated plasmid carrying the Ter sites. This response was shown to be dependent on ATR using the ATR inhibitor VE-822. This contrasts with a single Cas9-induced DSB between the two Ter sites, which causes a more spread gammaH2AX response. While this was monitored only at a single distal site, the difference between the DSB and the Tus-induced stall is very significant. Interestingly, despite evidence for ATR activation through the gammaH2AX response, no evidence for phosphorylation of ATR-T1989, CHK1-S345, or RPA2-S33 could be found under fork stalling conditions. The global replication inhibitor hydroxyurea (HU) elicited phosphorylation of ATR-T1989, CHK1-S345, or RPA2-S33. In this context, it would have been of interest to examine if a single DSB in the Ter region leads to phosphorylation of ATR-T1989, CHK1-S345, or RPA2-S33 and cell cycle arrest. It is not shown whether the replication inhibitor HU leads to the same widely spread gamma H2AX response. Overall, this is a well written manuscript, and the data provide convincing evidence that the Tus-Ter system poses a site-specific replication fork block in MCF7 cells leading to a localized ATR-dependent DNA damage checkpoint response that is distinct from the more global response to HU or DSBs.

Author response to public review:

“It is unclear whether the Ter sites integrated by a single copy plasmid have any effect on the replication of this region but the data show that the observed effects are dependent on expression of the Tus protein.”

-The lack of perturbation of the TerB sequence on fork progression has extensively been studied previously in both Willis et al, 2014 and Larsen et. al, 2014. Furthermore, as the detection of the SMARD signal at the TerB sites is dependent on the 7.5kb probe that spans the TerB sites (orange probe, Fig 2B & 2D), it would be impossible to study the effect on replication in this region, with and without the integration of the single copy plasmid.

“The SMARD data do not reveal what proportion of forks are arrested at Tus/Ter, or how long the fork delay is imposed.”

-The percentage of fork stalling at the TerB sites, with and without Tus expression, has been quantified in Figure 2E & 2F. Essentially, 36% forks stall at the TerB block, i.e. 18% of the forks stall in both the 5’ to 3’ (orange) and 3’ to 5’ (blue) direction when the Tus-TerB block is active.

“It is not shown whether the replication inhibitor HU leads to the same widely spread gamma H2AX response.”

-While we have not shown gH2AX accumulation via ChIP after HU treatment, Supplementary Figure 5A & 5B clearly show increased gH2AX foci when the cells are treated with HU, suggesting a global replication stress response that is in stark contrast to the response to Tus-TerB.

**Recommendations for the authors:**
Lines 78, 95: In the experimental set-up there are two divergent 5-TerB sites in the orientation that is non-permissive for the fork progression notwithstanding the direction. This raises an obvious question: How an intervening (~1kb-long) DNA segment in being replicated? Does it stay under-replicated and then break?

-The reviewers pose an important question about how the intervening sequence flanked by the two TerB sites is replicated, and if this leads to formation of anaphase bridges resulting in breaks. We think this is very plausible and this very question is part of ongoing studies in the lab with the aim to understand how the cell resolves a site-specific block. Unfortunately, this falls outside the scope of the current study.

Also, it is unclear what is meant with non-permissive orientation. This depends on the predominant replication direction. As the construct has Ter repeats in opposite orientation, any direction is non-permissive. These descriptions could be rephrased to avoid confusion

-The text has been edited to clarify this.

Fig 1A: It would be helpful to annotate the map to show the position of each primer relative to the Ter array. Why is there no signal for pp52?

-Figure 1A has the map of the locus with the annotated primer pairs and their relative positions to the TerB array.

-pp52 is positioned beyond the TerB array so binding of the Tus-His protein there is unlikely, confirming the specificity of the Tus binding to only the TerB array and not to the adjacent chromatin.

Figure 1B: Change Tus to Tus-His to make it easier to understand that the anti-His ChIP is targeting Tus. Provide information what normalization method was used in the ChIP experiments.

-Figure 1B has been edited to reflect this change

Line 113: Willis et al. 2014 also worked with chromosomal Ter sites, which should be acknowledged here.

The text has been modified to indicate this. We apologize for the oversight.

Line 126: Define pWB15 and its significance in text.

-The text has been edited to clarify this and mentions pWB15.

Figure 2E, F: Define legend (blue, orange boxes and arrow heads).

-The figure legend corresponding to Figure 2 has a detailed description of the boxes and the arrows.

Figure 3E, 4C: Add map of primers like in Figures 1 and 2.

-The map added to Figures 3 & 4 and text updated.

Figure 4: Showing that the gammaH2AX response is spread like with the single DSB would bolster the conclusion about the difference between a local and global response. Fig 4A, Lane-3: A loading control for the chromatin fraction is missing.

-Measuring gH2AX chromatin spread after global replication stress can be challenging. We have tried to address the question of global and local gH2AX response post replication stress by quantifying gH2AX foci in cells treated with and without hydroxyurea, comparing it with cells that have a functional Tus-TerB block (Supplementary Figure 5A& 5B). A single fork block seems to only elicit a local response while a global replication stress leads to gH2AX accumulation globally in the cell.

-Lamin A/C has been added to Fig 4A as a loading control for the chromatin fraction.

Figure S4: Analyzing ATR, CHK1 and RPA phosphorylation as well as cell cycle profile under single DSB condition may reveal that different localized responses exist. I mention this because it was reported in yeast that a single DSB in G1 cells leads to a similarly localized Mec1 (ATR) -dependent response that does not elicit phosphorylation of Rad53 (CHK1) and other downstream targets, but leads to H2A phosphorylation as well as phosphorylation of RPA and the Rad51 paralog Rad55 (see PMCID: PMC2853130). It might be of interest to the reader to discuss this publication and the commonalities and differences between both localized checkpoint response

-The reviewers raise an interesting question about the phosphorylation of ATR/CHK1/RPA and its effect on cell cycle after a single DSB. The aim of using the Cas9 break site in this study was merely to corroborate previously published observations pertaining to the spread of gH2AX after a DSB and to contrast that with the local response seen with Tus-TerB. Thus, while an intriguing question, we do not think this particular experiment will help in the understanding of the localized checkpoint response after a single replication fork block. However, we have included the observations previous published in the yeast system (PMC2853130) in our discussion as it helps compare and contrast fork blocks and DSBs further. It is of worth though that the yeast studies were looking at the cellular response to a DSB in G1.

Lines 256-260: In the discussion of ATRIP, unpublished data are discussed that show no increase in ssDNA. What is the effect of ATRIP depletion? Maybe delete this mention of unpublished data, if no new data can be provided. The authors are aware that this makes the mechanism of ATR activation at the 5-TerB site elusive.

-This statement has been deleted and the text has been modified.

Another possibility discussed by the authors is fork reversal. Since Tus/Ter complex block the CMG progression, fork reversal would result in a chicken foot structure with the long single-stranded 3'-overhang of an Okazaki fragment site. Such a structure should be protected by BRCA2 or RAD52 proteins from degradation. Any role for these proteins in the checkpoint activation at the TerB site?

-The reviewers suggest an interesting scenario where the Tus-TerB block induced reversed fork structure could be protected by the loading of known DNA repair proteins and this in turn could lead to a signaling mechanism and checkpoint activation. While we have not tested this hypothesis, nor studied the temporal dynamics of the formation if the reversed fork with respect to gH2AX accumulation, we think the localized gH2AX signal observed in the vicinity of the block is what initiates the downstream DDR response, promoting fork stabilization, followed either by fork reversal and restart or fork collapse. If the reversed fork was responsible for the gH2AX signaling, one would envision the spread to be more widespread, perhaps decorating the entire stretch of DNA between the block and the reversed fork. However, further studies are warranted to tease out this mechanism and the spatio-temporal dynamics.

Lines 292-294: The authors state that "unpublished work from our laboratory has demonstrated that replication forks are cleaved at or near the TerB site..." Unless the data are shown, it might be best to eliminate discussion of unpublished work, also because the occurrence of DNA ends at Ter sites was already described in Willis et al. 2017.

-The statement has been deleted and Willis et al. 2017 has been referenced.

Suppl Table 1: It would help to also show representative images of stretched fibers in addition to the summary data shown.

-Since the data is negative, the fiber images do not show any discernible differences and we do not think it adds useful information.

Suppl Fig 4. ChIP for gamma H2AX data. It would be helpful to show the distribution of the gamma H2AX signal along the chromosome for both the DSB response and the Tus/Ter response.

-The gH2AX ChIP signal at PP0-2 and PP10 has been included in Supplementary Fig4D. Though not significant for PP0-2, the data strongly suggests that there is increased spread of gH2AX along the chromosome after a DSB, strongly contrasting with the response after Tus-TerB block. The text has been modified to include both primer pairs.